# Distributed Finite-time Prescribed Performance Security Control for Unmanned Ships Utilizing the Novel Disturbance Estimator

Yuhui Song
*The School of Mathematical Sciences*
*Bohai University*
Jinzhou, Liaoning, China
e-mail: 18342628687@163.com

Huanqing Wang
*The School of Mathematical Sciences*
*Bohai University*
Jinzhou, Liaoning, China
e-mail: ndwhq@163.com

Siwen Liu
*The Navigation College*
*Dalian Maritime University*
Dalian, Liaoning, China
e-mail: SiwenLiu0220@163.com

Tieshan Li
*The School of Automation Engineering*
*University of Electronic Science*
*and Technology of China*
Chengdu, Sichuan, China
e-mail: tieshanli@126.com

*Abstract*—The article developed a distributed security control protocol of unmanned ships (USs) with the disturbances. To constraint the tracking errors within a predefined boundary, an infinite-time performance function and a logarithmic Lyapunov function are simultaneously introduced. Using the novel disturbance estimator, the disturbances are estimated. With the aid of the backstepping method, a distributed security controller is obtained and it is given that all the states of the system are bounded in finite time and the tracking errors can also meet preset performance requirements. Eventually, unmanned ships are given to show the utilizability of the obtained protocol.

*Index Terms*—finite-time control, security control, disturbance estimator, unmanned ships (USs)

## I. INTRODUCTION

USs are primarily preferred for tasks characterized by being dangerous or unsuitable for manned vessels. In the future, they will be developed for ocean surveying, meteorological monitoring, maritime search and rescue, and more. Due to its excellent characteristics, USs have been listed as a key research object in the field of marine armaments by countries around the world. In order to deal with the US control issue effectively, some control strategies were proposed [1]–[3]. Recently, the cooperative control for USs has attracted more and more attention due to that multiple USs can achieve the cooperative tasks efficiently [4]–[6]. In general, the cooperative control problem includes the tracking issue and the regulation issue. With regard to the control issue, all the USs will eventually incline to a non-specified value that is not previously known. Then, the follower USs can incline to the reference formed by the leader US or the convex hull

This research was supported by the National Natural Science Foundation of China under Grant No. 51939001, No. 61976033 and No. 62173046.

generated by the trajectories of the leader USs [7], [8]. When there exist multiple leader USs, that is called the containment control. As a matter of fact, it is invulnerable for multiple USs if the follower USs incline to the convex hull generated by the leader USs [9].

Although the backstepping control technique has become one of the most common methods to overcome the control issue of nonlinear systems [10]–[12], it is not difficult to obtain that, utilizing the classical backstepping technique to dispose the control issue of nonlinear systems, there exists the complexity explosion problem that is produced by repeated differentiation of virtual control variables. This may lead to the rapid multiplication of the systems and further makes the system performance unsatisfactory or even unstable [13]. Even if the proposed controller makes the performance of the studied system satisfy our design requirements, it is clear that the actual controller will become more complex [14]. It is obvious that the complex controller structure limits the practicability of the proposed strategy. To solve the above difficulty, the dynamic surface control method (DSCM) was presented [15], such that, by the conventional backstepping control technique, the designed first-order filter can be applied to work out the intermediate control variable's derivative at each step. After that, the method was extended to handle the control issue of different nonlinear systems, especially, nonlinear multi-agent systems [16], [17].

However, in the above articles, the infinite time boundedness of the states can be obtained. From the view of the convergence time, this may restrict the use of the developed controllers. Therefore, in the article, the finite-time first-order filter is applied to work out the intermediate control variable's derivative and, employing the finite-time control way, the convergence

of all the variables of USs will also be assured in finite time. In reality, many real systems, such as temperature of chemical reactor and physical stoppages, are with the constraints of the output and such constraints may come from safety consideration. Thus, it is very necessary to consider the problem of output constraints during the engineering system analysis. In order to handle such issue, the prescribed performance control (PPC) was developed [18]–[22]. By using a PPC method, a tracking controller was given for USs [23], and the PPC issue was discussed for USs in [24].

Consequently, the article discuss the distributed security control issue of USs with the disturbances. The major innovations of this work are as follows:

(1) By utilizing the proposed finite-time disturbance estimator, the disturbances existing in USs can be approximated.

(2) In this paper, by introducing the infinite-time performance function, a finite-time distributed security control scheme is obtained for USs with multiple leader USs.

The following paper structures are shown. Section II shows the problem formulation and preliminaries. Main results are shown in Section III. In Section IV, the simulation examples will prove the utilizability of the obtained method. The conclusion is shown in Section V.

## II. PRELIMINARIES AND PROBLEM FORMULATION

### A. US dynamics

Consider $M$ USs including $N$ follower USs and $M - N$ leader USs. The model of $i$th follower US is [25]

$$\begin{cases} \dot{\eta}_i = J_i(\psi_i)v_i \\ M\dot{v}_i + Dv_i = \tau_i + d_i(t) \end{cases} \quad (1)$$

where $\eta_i = [x_i, y_i, \psi_i]^T$ denote the position and direction. $v_i = [u_i, v_i, r_i]^T$ is the velocity. $M$ is the inertia matrix, $D$ is the damping matrix, and $\tau_i = [\tau_{iu}, \tau_{iv}, \tau_{ir}]^T$ is the control vector. $d_i(t)$ is the disturbance. It is assumed that the external disturbances and the derivatives of them satisfy $|\tau_{iu}| \leq \bar{\tau}_{iu}$, $|\tau_{iv}| \leq \bar{\tau}_{iv}$, $|\tau_{ir}| \leq \bar{\tau}_{ir}$, $|\dot{\tau}_{iu}| \leq \bar{\dot{\tau}}_{iu}$, $|\dot{\tau}_{iv}| \leq \bar{\dot{\tau}}_{iv}$, $|\dot{\tau}_{ir}| \leq \bar{\dot{\tau}}_{ir}$, where $\bar{\tau}_{iu}$, $\bar{\tau}_{iv}$, $\bar{\tau}_{ir}$, $\bar{\dot{\tau}}_{iu}$, $\bar{\dot{\tau}}_{iv}$ and $\bar{\dot{\tau}}_{ir}$ represent positive constants. $J_i(\psi_i)$ is the rotation matrix described as $J_i(\psi_i) = [\cos(\psi_i), -\sin(\psi_i), 0; \sin(\psi_i), \cos(\psi_i), 0; 0, 0, 1]$.

Then, define the reference of the $j$th leader US as $\eta_{jd} = \{x_{jd}, y_{jd}, \psi_{jd}\}$, where $j = N + 1, N + 2, \ldots, M$.

**Lemma 1** [26]: For the system $\dot{x} = f(x)$, if there is a continuous function $V(x)$ such that $\dot{V}(x) \leq -\lambda_1 V(x) - \lambda_2 V^\gamma(x) + \eta$, where $\lambda_1 > 0$, $\lambda_2 > 0$, $0 < \gamma < 1$, $0 < \eta < \infty$, then the trajectory of the system $\dot{x} = f(x)$ is practical finite-time stable, and the residual set is $\lim_{t \to T_r} V(x) \leq \min\{\frac{\eta}{(1-\theta_0)\lambda_1}, (\frac{\eta}{(1-\theta_0)\lambda_2})^{\frac{1}{\gamma}}\}$ where $\theta_0$ satisfies $0 < \theta_0 < 1$. The setting time is bounded as $T_r \leq \max\{t_0 + \frac{1}{\theta_0\lambda_1(1-\gamma)} \ln \frac{\theta_0\lambda_1 V^{1-\gamma}(t_0)+\lambda_2}{\lambda_2}, t_0 + \frac{1}{\lambda_1(1-\gamma)} \ln \frac{\lambda_1 V^{1-\gamma}(t_0)+\theta_0\lambda_2}{\theta_0\lambda_2}\}$.

**Lemma 2** [27]: Considering $\chi_i \in \mathcal{R}(i = 1, \ldots, n)$ and $\kappa \in [0, 1]$, one gets $(|\chi_1| + \cdots + |\chi_n|)^\kappa \leq |\chi_1|^\kappa + \cdots + |\chi_n|^\kappa$.

**Lemma 3** [28]: For $\Xi > 0$ and $x \in R$, it gets $0 \leq |x| - x \cdot \tanh(\frac{x}{\Xi}) \leq \kappa\Xi$ with $\kappa = \sup_{t>0}\left(\frac{1}{1+e^t}\right) = 0.2785$.

**Lemma 4** [29]: Define $\Omega_e = \{e||e| < 0.8814\bar{v}\}$ where $\bar{v}$ is a positive constant. If any $e \notin \Omega_e$, we can obtain $1 - 2\tanh^2(\frac{e}{\bar{v}}) \leq 0$.

### B. Basic graph theory

The information flow among $M$ USs including $N$ follower USs and $M - N$ leader USs is described by $G = (\Upsilon, E, A)$, where $\Upsilon = \{r_1, r_2, \ldots, r_m\}$ is the set of USs, $E \subseteq \Upsilon \times \Upsilon$ is the set of edges, and $A = [a_{ij}] \in R^{M \times M}$ is the weighted adjacency matrix. $r_i$ is the $i$th US and a directed edge $(i, j)$ means that US$\sharp i$ can obtain information from US$\sharp j$, i.e., $a_{ij} > 0$, otherwise, $a_{ij} = 0$ and define $a_{ii} = 0$. The set of neighbors of US$\sharp i$ is denoted as $\mathcal{N}_i = \{j|(j, i) \in E\}$. The Laplacian matrix is defined as $L_a = \mathscr{D} - A \in R^{M \times M}$ where $\mathscr{D} = \text{diag}[\mathscr{D}_1, \mathscr{D}_2, \ldots, \mathscr{D}_M]$, $\mathscr{D}_i = \Sigma_{j=1, j\neq i}^M a_{ij}$, $i = 1, 2, \ldots, M$. Since each follower US has at least one neighbor, and the leader USs have no neighbors, $L_a$ can be represented as [30] $L_a = [L_{1a}, L_{2a}; 0_{M \times N}, 0_{M \times M}]$ where $L_{1a} \in R^{N \times N}$ with $N$ denoting the number of follower USs and $L_{2a} \in R^{N \times M}$ with $M$ denoting the number of leader USs.

The containment control scheme is designed such that the position and direction of all follower USs converge into the convex hulls $Co(\mathbf{X}_d)$, $Co(\mathbf{Y}_d)$ and $Co(\psi_d)$, respectively, which are given by $Co(\mathbf{X}_d) = \{\Sigma_{N+1}^M c_i x_{id}|c_i \geq 0, \Sigma_{N+1}^M c_i = 1\}$, $Co(\mathbf{Y}_d) = \{\Sigma_{N+1}^M c_i y_{id}|c_i \geq 0, \Sigma_{N+1}^M c_i = 1\}$, $Co(\psi_d) = \{\Sigma_{N+1}^M c_i \psi_{id}|c_i \geq 0, \Sigma_{N+1}^M c_i = 1\}$ where $\mathbf{X}_d = \{x_{(N+1)d}, x_{(N+2)d}, \ldots, x_{Md}\}$, $\mathbf{y}_d = \{y_{(N+1)d}, y_{(N+2)d}, \ldots, y_{Md}\}$ and $\psi_d = \{\psi_{(N+1)d}, \psi_{(N+2)d}, \ldots, \psi_{Md}\}$.

Then the output synchronization error of the $i$th follower US is

$$\begin{cases} e_{i11} = \Sigma_{j=1}^N a_{ij}(x_i - x_j) + \Sigma_{j=N+1}^M a_{ij}(x_i - x_{jd}) \\ e_{i12} = \Sigma_{j=1}^N a_{ij}(y_i - y_j) + \Sigma_{j=N+1}^M a_{ij}(y_i - y_{jd}) \\ e_{i13} = \Sigma_{j=1}^N a_{ij}(\psi_i - \psi_j) + \Sigma_{j=N+1}^M a_{ij}(\psi_i - \psi_{jd}) \end{cases} \quad (2)$$

### C. Prescribed performance control

The PPC design is that the tracking error is included in the preset residual set, such that the tracking error $e_{i1j}, i = 1, 2, \ldots, N, j = 1, 2, 3$, is included in a predefined range, which is shown as

$$-v_{ij}(t) < e_{i1k}(t) < v_{ij}(t), \quad (3)$$

where $i = 1, 2, \ldots, N$, $j = x, y, \psi$, $k = 1, 2, 3$, $v_{ij}(t)$ satisfies $\lim_{t\to\infty} v_{ij}(t) = v_{tfij}$ and $v_{tfij}$ is the steady-state value of $e_{i1k}$.

Choose $v_{ij}(t) = (v_{0ij} - v_{tfij})e^{-\beta_{ij}t} + v_{tfij}$ as the performance function, where $v_{0ij}$, $v_{tfij}$ and $\beta_{ij}$ are positive constants and $v_{0ij} > v_{tfij}$. $v_{0ij} = v_{ij}(0)$ is chosen such that $-v_{ij}(0) < e_{i1k}(0) < v_{ij}(0)$. $\beta_{ij}$ represents the convergence rate of $e_{i1k}(t)$. Hence, by choosing $v_{0ij}$, $v_{tfij}$ and $\beta_{ij}$ to make the tracking error be included in the predefined range (3), the PPC can be obtained.

For the PPC design, the following lemma is shown.

**Lemma 5** [31]: For any positive function $v(t)$, we can know that the following inequality is established, when $e(t)$ remains in the interval $|e(t)| < v(t)$: $\ln \frac{v^2(t)}{v^2(t)-e^2(t)} < \frac{e^2(t)}{v^2(t)-e^2(t)}$.

## III. MAIN RESULTS

### A. The Design of Disturbance Estimator

In this paper, they are assumed that $M = [m_{11}, 0, 0; 0, m_{22}, 0; 0, 0, m_{33}]$ and $D = [d_{11}, 0, 0; 0, d_{22}, d_{23}; 0, d_{32}, d_{33}]$. Then, substituting $M$ and $D$ into (1) can yield $m_{11}\dot{u}_i = -d_{11}u_i + \tau_{iu} + d_{iu}, m_{22}\dot{v}_i = -d_{22}v_i - d_{23}r_i + \tau_{iv} + d_{iv}, m_{33}\dot{r}_i = -d_{33}r_i - d_{32}v_i + \tau_{ir} + d_{ir}$. Then, we create the following disturbance estimator.

$$
\begin{cases}
m_{11}\dot{\hat{u}}_i = -d_{11}u_i + \tau_{iu} + \hat{d}_{iu} \\
\quad\quad +k_{iu1}\tanh(\frac{u_i-\hat{u}_i}{\Xi_{\tilde{u}_i}}) + k_{iu2}(u_i - \hat{u}_i) \\
\dot{\hat{d}}_{iu} = \beta_{iu}(e_{i21} + u_i - \hat{u}_i) - \delta_{iu}\hat{d}_{iu} \\
m_{22}\dot{\hat{v}}_i = -d_{22}v_i - d_{23}r_i + \tau_{iv} + \hat{d}_{iv} \\
\quad\quad +k_{iv1}\tanh(\frac{v_i-\hat{v}_i}{\Xi_{\tilde{v}_i}}) + k_{iv2}(v_i - \hat{v}_i) \\
\dot{\hat{d}}_{iv} = \beta_{iv}(e_{i22} + v_i - \hat{v}_i) - \delta_{iv}\hat{d}_{iv} \\
m_{33}\dot{\hat{r}}_i = -d_{33}r_i - d_{32}v_i + \tau_{ir} + \hat{d}_{ir} \\
\quad\quad +k_{ir1}\tanh(\frac{r_i-\hat{r}_i}{\Xi_{\tilde{r}_i}}) + k_{ir2}(r_i - \hat{r}_i) \\
\dot{\hat{d}}_{ir} = \beta_{ir}(e_{i23} + r_i - \hat{r}_i) - \delta_{ir}\hat{d}_{ir}
\end{cases}
\tag{4}
$$

where $k_{iu1}$, $\Xi_{\tilde{u}_i}$, $k_{iu2}$, $\beta_{iu}$, $\delta_{iu}$, $k_{iv1}$, $\Xi_{\tilde{v}_i}$, $k_{iv2}$, $\beta_{iv}$, $\delta_{iv}$, $k_{ir1}$, $\Xi_{\tilde{r}_i}$, $k_{ir2}$, $\beta_{ir}$ and $\delta_{ir}$ are positive constants; $e_{i21}$, $e_{i22}$ and $e_{i23}$ will be defined later.

By defining $\tilde{u}_i = u_i - \hat{u}_i$, $\tilde{v}_i = v_i - \hat{v}_i$ and $\tilde{r}_i = r_i - \hat{r}_i$, we can obtain

$$
\begin{cases}
m_{11}\tilde{u}_i\dot{\tilde{u}}_i = \tilde{u}_i\tilde{d}_{iu} - k_{iu1}\tilde{u}_i\tanh(\frac{\tilde{u}_i}{\Xi_{\tilde{u}_i}}) - k_{iu2}\tilde{u}_i^2 \\
m_{22}\tilde{v}_i\dot{\tilde{v}}_i = \tilde{v}_i\tilde{d}_{iv} - k_{iv1}\tilde{v}_i\tanh(\frac{\tilde{v}_i}{\Xi_{\tilde{v}_i}}) - k_{iv2}\tilde{v}_i^2 \\
m_{33}\tilde{r}_i\dot{\tilde{r}}_i = \tilde{r}_i\tilde{d}_{ir} - k_{ir1}\tilde{r}_i\tanh(\frac{\tilde{r}_i}{\Xi_{\tilde{r}_i}}) - k_{ir2}\tilde{r}_i^2
\end{cases}
\tag{5}
$$

### B. The controller design process

We introduce the changes of coordinates as follows:

$$
\begin{cases}
e_{i21} = u_i - v_{idu}^d \\
e_{i22} = v_i - v_{idv}^d \\
e_{i23} = r_i - v_{idr}^d, i = 1, 2, \ldots, N
\end{cases}
\tag{6}
$$

where $v_{idj}^d$, the first-order filter output signal, is given as

$$
\dot{v}_{idj}^d = -k_{idj1}\tanh(\frac{v_{idj}^d - v_{idj}}{\Xi_{y_{idj}}}) - k_{idj2}(v_{idj}^d - v_{idj})
\tag{7}
$$

where $v_{idj}^d(0) = v_{idj}(0)$, $j = u, v, r$. $v_{idj}$ is the input. $k_{idj1}$, $\Xi_{y_{idj}}$ and $k_{idj2}$ are the design constants. The filter errors are $y_{idj} = v_{idj}^d - v_{idj}, j = u, v, r$.

**Step** 1. Firstly, considering the first equation in (2), we can get $\dot{e}_{i11} = \Sigma_{j=1}^M a_{ij}(e_{i21}\cos(\psi_i) - e_{i22}\sin(\psi_i)) + \Sigma_{j=1}^M a_{ij}(v_{idu}\cos(\psi_i) - v_{idv}\sin(\psi_i)) - \Sigma_{j=1}^N a_{ij}\dot{x}_j - \Sigma_{j=N+1}^M a_{ij}\dot{x}_{id} + \Sigma_{j=1}^M a_{ij}(y_{idu}\cos(\psi_i) - y_{idv}\sin(\varphi_i))$. Next,

the Lyapunov function is $V_{e_{i11}} = \frac{1}{2}\ln\frac{v_{ix}^2}{v_{ix}^2 - e_{i11}^2}$. Subsequently, the dynamics of $V_{e_{i11}}$ are

$$
\begin{aligned}
\dot{V}_{e_{i11}} &= \frac{e_{i21}e_{i11}\sum_{j=1}^M a_{ij}\cos(\psi_i)}{v_{ix}^2 - e_{i11}^2} \\
&\quad -\frac{e_{i22}e_{i11}\sum_{j=1}^M a_{ij}\sin(\psi_i)}{v_{ix}^2 - e_{i11}^2} \\
&\quad +\frac{e_{i11}}{v_{ix}^2 - e_{i11}^2}(\sum_{j=1}^M a_{ij}(v_{idu}\cos(\psi_i) - v_{idv}\sin(\psi_i) \\
&\quad -\sum_{j=1}^N a_{ij}\dot{x}_j - \sum_{j=N+1}^M a_{ij}\dot{x}_{id} \\
&\quad +\frac{2(v_{ix}^2 - e_{i11}^2)}{e_{i11}}\tanh^2(\frac{e_{i11}}{\bar{v}_{ix}})(-\frac{v_{ix}\dot{v}_{ix}}{v_{ix}^2 - e_{i11}^2}) \\
&\quad +\frac{e_{i11}}{v_{ix}^2 - e_{i11}^2}) + \frac{\dot{v}_{ix}}{v_{ix}} + (1 - 2\tanh^2(\frac{e_{i11}}{\bar{v}_{ix}}))(-\frac{v_{ix}\dot{v}_{ix}}{v_{ix}^2 - e_{i11}^2}) \\
&\quad +\frac{(\sum_{j=1}^M a_{ij})^2}{2}y_{idu}^2 + \frac{(\sum_{j=1}^M a_{ij})^2}{2}y_{idv}^2
\end{aligned}
\tag{8}
$$

Then, we design the following expression.

$$
\begin{aligned}
T_{ix} &= \frac{1}{\sum_{j=1}^M a_{ij}}(-k_{i1x}e_{i11} + \sum_{j=1}^N a_{ij}\dot{x}_j \\
&\quad +\sum_{j=N+1}^M a_{ij}\dot{x}_{id} - \frac{2(v_{ix}^2 - e_{i11}^2)}{e_{i11}}\tanh^2(\frac{e_{i11}}{\bar{v}_{ix}}) \\
&\quad \times(-\frac{v_{ix}\dot{v}_{ix}}{v_{ix}^2 - e_{i11}^2}) + \frac{e_{i11}}{v_{ix}^2 - e_{i11}^2})
\end{aligned}
\tag{9}
$$

where $T_{ix} = v_{idu}\cos(\psi_i) - v_{div}\sin(\psi_i)$ and $k_{i1x} > 0$.

Substituting (9) into (8), we have

$$
\begin{aligned}
\dot{V}_{e_{i11}} &= \frac{e_{i21}e_{i11}\sum_{j=1}^M a_{ij}\cos(\psi_i)}{v_{ix}^2 - e_{i11}^2} - \frac{e_{i22}e_{i11}\sum_{j=1}^M a_{ij}\sin(\psi_i)}{v_{ix}^2 - e_{i11}^2} \\
&\quad -k_{i1x}\frac{e_{i11}^2}{v_{ix}^2 - e_{i11}^2} + \frac{\dot{v}_{ix}}{v_{ix}} + (1 - 2\tanh^2(\frac{e_{i11}}{\bar{v}_{ix}}))(-\frac{v_{ix}\dot{v}_{ix}}{v_{ix}^2 - e_{i11}^2}) \\
&\quad +\frac{(\sum_{j=1}^M a_{ij})^2}{2}y_{idu}^2 + \frac{(\sum_{j=1}^M a_{ij})^2}{2}y_{idv}^2
\end{aligned}
\tag{10}
$$

Secondly, considering the second equation in (2), we can get

$$
\begin{aligned}
\dot{e}_{i12} &= \sum_{j=1}^M a_{ij}(e_{i21}\sin(\psi_i) + e_{i22}\cos(\psi_i)) + \sum_{j=1}^M a_{ij}(v_{idu}\sin(\psi_i) \\
&\quad +v_{idv}\cos(\psi_i)) - \sum_{j=1}^N a_{ij}\dot{y}_j - \sum_{j=N+1}^M a_{ij}\dot{y}_{id} \\
&\quad +\sum_{j=1}^M a_{ij}(y_{idu}\sin(\psi_i) + y_{idv}\cos(\psi_i))
\end{aligned}
\tag{11}
$$

The following Lyapunov function candidate is defined as $V_{ei12} = \frac{1}{2} \ln \frac{v_{iy}^2}{v_{iy}^2 - e_{i12}^2}$. Subsequently, the dynamics of $V_{ei12}$ are

$$\dot{V}_{ei12} = \frac{e_{i21}e_{i12}\sum_{j=1}^{M} a_{ij}\sin(\psi_i)}{v_{iy}^2 - e_{i12}^2} + \frac{e_{i22}e_{i12}\sum_{j=1}^{M} a_{ij}\cos(\psi_i)}{v_{iy}^2 - e_{i12}^2}$$

$$+\frac{e_{i12}}{v_{iy}^2 - e_{i12}^2}(\sum_{j=1}^{M} a_{ij}(v_{idu}\sin(\psi_i) + v_{idv}\cos(\psi_i)) - \sum_{j=1}^{N} a_{ij}\dot{y}_j$$

$$-\sum_{j=N+1}^{M} a_{ij}\dot{y}_{id} + \frac{2(v_{iy}^2 - e_{i12}^2)}{e_{i12}}\tanh^2(\frac{e_{i12}}{\bar{v}_{iy}})(-\frac{v_{iy}\dot{v}_{iy}}{v_{iy}^2 - e_{i12}^2})$$

$$+\frac{e_{i12}}{v_{iy}^2 - e_{i12}^2}) + \frac{\dot{v}_{iy}}{v_{iy}} + (1 - 2\tanh^2(\frac{e_{i12}}{\bar{v}_{iy}}))(-\frac{v_{iy}\dot{v}_{iy}}{v_{iy}^2 - e_{i12}^2})$$

$$+\frac{(\sum_{j=1}^{M} a_{ij})^2}{2}y_{idu}^2 + \frac{(\sum_{j=1}^{M} a_{ij})^2}{2}y_{idv}^2 \tag{12}$$

Then, we give the equation as follows:

$$T_{iy} = \frac{1}{\sum_{j=1}^{M} a_{ij}}(-k_{i1y}e_{i12} + \sum_{j=1}^{N} a_{ij}\dot{y}_j + \sum_{j=N+1}^{M} a_{ij}\dot{y}_{id}$$

$$-\frac{2(v_{iy}^2 - e_{i12}^2)}{e_{i12}}\tanh^2(\frac{e_{i12}}{\bar{v}_{iy}})(-\frac{v_{iy}\dot{v}_{iy}}{v_{iy}^2 - e_{i12}^2})$$

$$+\frac{e_{i12}}{v_{iy}^2 - e_{i12}^2}) \tag{13}$$

where $T_{iy} = v_{idu}\sin(\psi_i) + v_{div}\cos(\psi_i)$ and $k_{i1y} > 0$.

Substituting (13) into (12), we have

$$\dot{V}_{ei12} = \frac{e_{i21}e_{i21}\sum_{j=1}^{M} a_{ij}\sin(\psi_i)}{v_{iy}^2 - e_{i12}^2} + \frac{e_{i22}e_{i12}\sum_{j=1}^{M} a_{ij}\cos(\psi_i)}{v_{iy}^2 - e_{i12}^2}$$

$$-k_{i1y}\frac{e_{i12}^2}{v_{iy}^2 - e_{i12}^2} + \frac{\dot{v}_{iy}}{v_{iy}} + (1 - 2\tanh^2(\frac{e_{i12}}{\bar{v}_{iy}}))(-\frac{v_{iy}\dot{v}_{iy}}{v_{iy}^2 - e_{i12}^2})$$

$$+\frac{(\sum_{j=1}^{M} a_{ij})^2}{2}y_{idu}^2 + \frac{(\sum_{j=1}^{M} a_{ij})^2}{2}y_{idv}^2 \tag{14}$$

Finally, considering the third equation in (2), we can get $\dot{e}_{i13} = \Sigma_{j=1}^{M} a_{ij}(e_{i23} + v_{idr} + y_{idr}) - \Sigma_{j=1}^{N} a_{ij}\dot{\psi}_j - \Sigma_{j=N+1}^{M} a_{ij}\dot{\psi}_{id}$. The Lyapunov function candidate is defined as follows: $V_{ei13} = \frac{1}{2} \ln \frac{v_{i\psi}^2}{v_{i\psi}^2 - e_{i13}^2}$ Subsequently, the dynamics of $V_{ei13}$ are

$$\dot{V}_{ei13} = \frac{1}{2}(\frac{v_{i\psi}^2 - e_{i13}^2}{v_{i\psi}^2}\frac{2v_{i\psi}\dot{v}_{i\psi}(v_{i\psi}^2 - e_{i13}^2)}{(v_{i\psi}^2 - e_{i13}^2)^2}$$

$$\frac{-v_{i\psi}^2(2v_{i\psi}\dot{v}_{i\psi} - 2e_{i13}\dot{e}_{i13})}{(v_{i\psi}^2 - e_{i13}^2)^2})$$

$$= e_{i23}\sum_{j=1}^{M} a_{ij}e_{i13} + \frac{e_{i13}}{v_{i\psi}^2 - e_{i13}^2}(\sum_{j=1}^{M} a_{ij}v_{idr} - \sum_{j=1}^{N} a_{ij}\dot{\psi}_j$$

$$-\sum_{j=N+1}^{M} a_{ij}\dot{\psi}_{id} + \frac{2(v_{i\psi}^2 - e_{i13}^2)}{e_{i13}}\tanh^2(\frac{e_{i13}}{\bar{v}_{i\psi}})(-\frac{v_{i\psi}\dot{v}_{i\psi}}{v_{i\psi}^2 - e_{i13}^2})$$

$$+\frac{1}{2}\frac{e_{i13}}{v_{i\psi}^2 - e_{i13}^2}) + \frac{\dot{v}_{i\psi}}{v_{i\psi}} + (1 - 2\tanh^2(\frac{e_{i13}}{\bar{v}_{i\psi}}))(-\frac{v_{i\psi}\dot{v}_{i\psi}}{v_{i\psi}^2 - e_{i13}^2})$$

$$+\frac{(\sum_{j=1}^{M} a_{ij})^2}{2}y_{idr}^2 \tag{15}$$

Then, we design the following equation.

$$T_{i\psi} = \frac{1}{\sum_{j=1}^{M} a_{ij}}(-k_{i1\psi}e_{i13} + \sum_{j=1}^{N} a_{ij}\dot{\psi}_j + \sum_{j=N+1}^{M} a_{ij}\dot{\psi}_{id}$$

$$-\frac{2(v_{i\psi}^2 - e_{i13}^2)}{e_{i13}}\tanh^2(\frac{e_{i13}}{\bar{v}_{i\psi}})(-\frac{v_{i\psi}\dot{v}_{i\psi}}{v_{i\psi}^2 - e_{i13}^2})$$

$$+\frac{1}{2}\frac{e_{i13}}{v_{i\psi}^2 - e_{i13}^2}) \tag{16}$$

where $T_{i\psi} = v_{idr}$ and $k_{i1\psi} > 0$.

Substituting (16) into (15), we have

$$\dot{V}_{ei13} = e_{i23} \cdot \sum_{j=1}^{M} a_{ij}e_{i13} - k_{i1\psi}\frac{e_{i13}^2}{v_{i\psi}^2 - e_{i13}^2} + \frac{\dot{v}_{i\psi}}{v_{i\psi}} + (1$$

$$-2\tanh^2(\frac{e_{i13}}{\bar{v}_{i\psi}}))(-\frac{v_{i\psi}\dot{v}_{i\psi}}{v_{i\psi}^2 - e_{i13}^2}) + \frac{(\sum_{j=1}^{M} a_{ij})^2}{2}y_{idr}^2 \tag{17}$$

Define $T_i = [T_{ix}, T_{iy}, T_{i\psi}]^T$. Then, according to (9), (13) and (16), we can obtain $v_{id} = J_i^{-1}(\psi_i)T_i$ where $v_{id} = [v_{idu}, v_{idv}, v_{idr}]^T$.

**Step** 2. Firstly, considering the first equation in (6), we can get $\dot{e}_{i21} = \dot{u}_i - \dot{v}_{idu}^d$. The following Lyapunov function candidate is defined as $V_{ei21} = \frac{1}{2}m_{11}e_{i21}^2 + \frac{1}{2}m_{11}\tilde{u}_i^2 + \frac{1}{2\beta_{iu}}\tilde{d}_{iu}^2 + \frac{1}{2}y_{idu}^2$ where $\tilde{d}_{iu} = d_{iu} - \hat{d}_{iu}$ and $\beta_{iu} > 0$.

Using (5) and (7), the dynamics of $V_{ei21}$ are

$$\dot{V}_{ei21} \leq e_{i21}(-d_{11}u_i + \tau_{iu} + \hat{d}_{iu} - m_{11}\dot{v}_{idu}^d$$

$$+\frac{e_{i11}\sum_{j=1}^{M} a_{ij}\cos(\psi_i)}{v_{ix}^2 - e_{i11}^2} + \frac{e_{i12}\sum_{j=1}^{M} a_{ij}\sin(\psi_i)}{v_{iy}^2 - e_{i12}^2})$$

$$-k_{iu1}\tilde{u}_i\tanh(\frac{\tilde{u}_i}{\Xi_{\tilde{u}_i}}) - k_{iu2}\tilde{u}_i^2 + \frac{1}{\beta_{iu}}\tilde{d}_{iu}\dot{d}_{iu} + \frac{1}{\beta_{iu}}\tilde{d}_{iu}(\beta_{iu}(e_{i21}$$

$$-\tilde{u}_i) - \dot{\hat{d}}_{iu}) - k_{idu1}y_{idu}\tanh(\frac{y_{idu}}{\Xi_{y_{idu}}}) - (k_{idu2} - \frac{1}{2})y_{idu}^2$$

$$+\frac{1}{2}(\dot{v}_{idu})^2 - e_{i21} \cdot \frac{e_{i11}\sum_{j=1}^{M} a_{ij}\cos(\psi_i)}{v_{ix}^2 - e_{i11}^2}$$

$$-e_{i21} \cdot \frac{e_{i12}\sum_{j=1}^{M} a_{ij}\sin(\psi_i)}{v_{iy}^2 - e_{i12}^2} \tag{18}$$

Subsequently, the force $\tau_{iu}$ in surge is designed as follows:

$$\tau_{iu} = -k_{i21u}\tanh(\frac{e_{i21}}{\Xi_{e_{i21}}}) - k_{i22u}e_{i21} + d_{11}u_i - \hat{d}_{iu}$$

$$+m_{11}\dot{v}_{idu}^d - \frac{e_{i11}\sum_{j=1}^{M} a_{ij}\cos(\psi_i)}{v_{ix}^2 - e_{i11}^2}$$

$$-\frac{e_{i12}\sum_{j=1}^{M} a_{ij}\sin(\psi_i)}{v_{iy}^2 - e_{i12}^2} \tag{19}$$

where $k_{i21u} > 0$ and $k_{i22u} > 0$.

Substituting (4) and (19) into (18), we have

$$\dot{V}_{e_{i21}} \leq -k_{i21u}e_{i21}\tanh(\frac{e_{i21}}{\Xi_{e_{i21}}}) - k_{i22u}e_{i21}^2 - k_{iu1}\tilde{u}_i\tanh(\frac{\tilde{u}_i}{\Xi_{\tilde{u}_i}})$$
$$-k_{iu2}\tilde{u}_i^2 + \frac{1}{\beta_{iu}}\tilde{d}_{iu}\dot{d}_{iu} + \frac{\delta_{iu}}{\beta_{iu}}\tilde{d}_{iu}\hat{d}_{iu} - k_{idu1}y_{idu}\tanh(\frac{y_{idu}}{\Xi_{y_{idu}}})$$
$$-(k_{idu2} - \frac{1}{2})y_{idu}^2 + \frac{1}{2}(\dot{\upsilon}_{idu})^2 - e_{i21}\cdot\frac{e_{i11}\sum_{j=1}^M a_{ij}\cos(\psi_i)}{v_{ix}^2 - e_{i11}^2}$$
$$-e_{i21}\cdot\frac{e_{i12}\sum_{j=1}^M a_{ij}\sin(\psi_i)}{v_{iy}^2 - e_{i12}^2} \tag{20}$$

Utilizing Lemma 3 and Young's inequality, we get

$$-k_{i21u}e_{i21}\tanh(\frac{e_{i21}}{\Xi_{e_{i21}}}) \leq -k_{i21u}(e_{i21}^2)^{\frac{1}{2}} + k_{i21u}\kappa\Xi_{e_{i21}}$$
$$-k_{iu1}\tilde{u}_i\tanh(\frac{\tilde{u}_i}{\Xi_{\tilde{u}_i}}) \leq -k_{iu1}(\tilde{u}_i^2)^{\frac{1}{2}} + k_{iu1}\kappa\Xi_{\tilde{u}_i}$$
$$-k_{idu1}y_{idu}\tanh(\frac{y_{idu}}{\Xi_{y_{idu}}}) \leq -k_{idu1}(y_{idu}^2)^{\frac{1}{2}} + k_{idu1}\kappa\Xi_{y_{idu}}$$
$$\frac{1}{\beta_{iu}}\tilde{d}_{iu}\dot{d}_{iu} + \frac{\delta_{iu}}{\beta_{iu}}\tilde{d}_{iu}\hat{d}_{iu} \leq -(\frac{1}{2\beta_{iu}}\tilde{d}_{iu}^2)^{\frac{1}{2}} - \frac{\delta_{iu}-2}{2\beta_{iu}}\tilde{d}_{iu}^2$$
$$+\frac{1}{2\beta_{iu}}(\dot{d}_{iu})^2 + \frac{\delta_{iu}}{2\beta_{iu}}d_{iu}^2$$
$$+\frac{1}{2} \tag{21}$$

Combining (20) and (21), we have

$$\dot{V}_{e_{i21}} \leq -k_{i21u}(e_{i21}^2)^{\frac{1}{2}} - k_{i22u}e_{i21}^2 - k_{iu1}(\tilde{u}_i^2)^{\frac{1}{2}} - k_{iu2}\tilde{u}_i^2$$
$$-(\frac{1}{2\beta_{iu}}\tilde{d}_{iu}^2)^{\frac{1}{2}} - \frac{\delta_{iu}-2}{2\beta_{iu}}\tilde{d}_{iu}^2 - k_{idu1}(y_{idu}^2)^{\frac{1}{2}} - (k_{idu2} - \frac{1}{2})y_{idu}^2$$
$$-e_{i21}\cdot\frac{e_{i11}\sum_{j=1}^M a_{ij}\cos(\psi_i)}{v_{ix}^2 - e_{i11}^2} - e_{i21}\cdot\frac{e_{i12}\sum_{j=1}^M a_{ij}\sin(\psi_i)}{v_{iy}^2 - e_{i12}^2}$$
$$+T_{e_{i21}} \tag{22}$$

where $T_{e_{i21}} = k_{i21u}\kappa\Xi_{e_{i21}} + k_{iu1}\kappa\Xi_{\tilde{u}_i} + k_{idu1}\kappa\Xi_{y_{idu}} + \frac{1}{2}(\dot{\upsilon}_{idu})^2 + \frac{1}{2\beta_{iu}}(\dot{d}_{iu})^2 + \frac{\delta_{iu}}{2\beta_{iu}}d_{iu}^2 + \frac{1}{2}$.

Secondly, considering the second equation in (6), we can get $\dot{e}_{i22} = \dot{v}_i - \dot{\upsilon}_{idv}^d$. The Lyapunov function candidate is defined as follows: $V_{e_{i22}} = \frac{1}{2}m_{22}e_{i22}^2 + \frac{1}{2}m_{22}\tilde{v}_i^2 + \frac{1}{2\beta_{iv}}\tilde{d}_{iv}^2 + \frac{1}{2}y_{idv}^2$ where $\tilde{d}_{iu} = d_{iu} - \hat{d}_{iu}$ and $\beta_{iu} > 0$. Using (5) and (7), the dynamics of $V_{e_{i22}}$ are

$$\dot{V}_{e_{i22}} \leq e_{i22}(-d_{22}v_i - d_{23}r_i + \tau_{iv} + \hat{d}_{iv} - m_{22}\dot{d}_{idv}^d$$
$$-\frac{e_{i11}\sum_{j=1}^M a_{ij}\sin(\psi_i)}{v_{ix}^2 - e_{i11}^2} + \frac{e_{i12}\sum_{j=1}^M a_{ij}\cos(\psi_i)}{v_{iy}^2 - e_{i12}^2})$$
$$-k_{iv1}\tilde{v}_i\tanh(\frac{\tilde{v}_i}{\Xi_{\tilde{v}_i}}) - k_{iv2}\tilde{v}_i^2 + \frac{1}{\beta_{iv}}\tilde{d}_{iv}\dot{d}_{iv} + \frac{1}{\beta_{iv}}\tilde{d}_{iv}(\beta_{iv}(e_{i22}$$
$$+\tilde{v}_i) - \dot{\hat{d}}_{iv}) - k_{idv1}y_{idv}\tanh(\frac{y_{idv}}{\Xi_{y_{idv}}}) - (k_{idv2} - \frac{1}{2})y_{idv}^2$$
$$+\frac{1}{2}(\dot{\upsilon}_{idv})^2 + e_{i22}\cdot\frac{e_{i11}\sum_{j=1}^M a_{ij}\sin(\psi_i)}{v_{ix}^2 - e_{i11}^2}$$
$$-e_{i22}\cdot\frac{e_{i12}\sum_{j=1}^M a_{ij}\cos(\psi_i)}{v_{iy}^2 - e_{i12}^2} \tag{23}$$

Subsequently, the force $\tau_{iv}$ in sway is designed as follows:

$$\tau_{iv} = -k_{i21v}\tanh(\frac{e_{i22}}{\Xi_{e_{i22}}}) - k_{i22v}e_{i22} + d_{22}v_i$$
$$+d_{23}r_i - \hat{d}_{iv} + m_{22}\dot{d}_{idv}^d + \frac{e_{i11}\sum_{j=1}^M a_{ij}\sin(\psi_i)}{v_{ix}^2 - e_{i11}^2}$$
$$-\frac{e_{i12}\sum_{j=1}^M a_{ij}\cos(\psi_i)}{v_{iy}^2 - e_{i12}^2} \tag{24}$$

where $k_{i21v} > 0$ and $k_{i22v} > 0$.

Substituting (4) and (24) into (23), we have

$$\dot{V}_{e_{i22}} \leq -k_{i21v}e_{i22}\tanh(\frac{e_{i22}}{\Xi_{e_{i22}}}) - k_{i22v}e_{i22}^2$$
$$-k_{iv1}\tilde{v}_i\tanh(\frac{\tilde{v}_i}{\Xi_{\tilde{v}_i}}) - k_{iv2}\tilde{v}_i^2 + \frac{1}{\beta_{iv}}\tilde{d}_{iv}\dot{d}_{iv} + \frac{\delta_{iv}}{\beta_{iv}}\tilde{d}_{iv}\hat{d}_{iv}$$
$$-k_{idv1}y_{idv}\tanh(\frac{y_{idv}}{\Xi_{y_{idv}}}) - (k_{idv2} - \frac{1}{2})y_{idv}^2 + \frac{1}{2}(\dot{\upsilon}_{idv})^2$$
$$+e_{i22}\cdot\frac{e_{i11}\sum_{j=1}^M a_{ij}\sin(\psi_i)}{v_{ix}^2 - e_{i11}^2}$$
$$-e_{i22}\cdot\frac{e_{i12}\sum_{j=1}^M a_{ij}\cos(\psi_i)}{v_{iy}^2 - e_{i12}^2} \tag{25}$$

Utilizing Lemma 3 and Young's inequality, we get

$$-k_{i21v}e_{i22}\tanh(\frac{e_{i22}}{\Xi_{e_{i22}}}) \leq -k_{i21v}(e_{i22}^2)^{\frac{1}{2}} + k_{i21v}\kappa\Xi_{e_{i22}}$$
$$-k_{iv1}\tilde{v}_i\tanh(\frac{\tilde{v}_i}{\Xi_{\tilde{v}_i}}) \leq -k_{iv1}(\tilde{v}_i^2)^{\frac{1}{2}} + k_{iv1}\kappa\Xi_{\tilde{v}_i}$$
$$-k_{idv1}y_{idv}\tanh(\frac{y_{idv}}{\Xi_{y_{idv}}}) \leq -k_{idv1}(y_{idv}^2)^{\frac{1}{2}} + k_{idv1}\kappa\Xi_{y_{idv}}$$
$$\frac{1}{\beta_{iv}}\tilde{d}_{iv}\dot{d}_{iv} + \frac{\delta_{iv}}{\beta_{iv}}\tilde{d}_{iv}\hat{d}_{iv} \leq -(\frac{1}{2\beta_{iv}}\tilde{d}_{iv}^2)^{\frac{1}{2}} - \frac{\delta_{iv}-2}{2\beta_{iv}}\tilde{d}_{iv}^2$$
$$+\frac{1}{2\beta_{iv}}(\dot{d}_{iv})^2 + \frac{\delta_{iv}}{2\beta_{iv}}d_{iv}^2$$
$$+\frac{1}{2} \tag{26}$$

Combining (25) and (26), we have

$$\dot{V}_{e_{i22}} \leq -k_{i21v}(e_{i22}^2)^{\frac{1}{2}} - k_{i22v}e_{i22}^2 - k_{iv1}(\tilde{v}_i^2)^{\frac{1}{2}} - k_{iv2}\tilde{v}_i^2$$
$$-(\frac{1}{2\beta_{iv}}\tilde{d}_{iv}^2)^{\frac{1}{2}} - \frac{\delta_{iv}-2}{2\beta_{iv}}\tilde{d}_{iv}^2 - k_{idv1}(y_{idv}^2)^{\frac{1}{2}} - (k_{idv2}$$
$$-\frac{1}{2})y_{idv}^2 + e_{i22}\cdot\frac{e_{i11}\sum_{j=1}^M a_{ij}\sin(\psi_i)}{v_{ix}^2 - e_{i11}^2}$$
$$-e_{i22}\cdot\frac{e_{i12}\sum_{j=1}^M a_{ij}\cos(\psi_i)}{v_{iy}^2 - e_{i12}^2} + T_{e_{i22}} \tag{27}$$

where $T_{e_{i22}} = k_{i21v}\kappa\Xi_{e_{i22}} + k_{iv1}\kappa\Xi_{\tilde{v}_i} + k_{idv1}\kappa\Xi_{y_{idv}} + \frac{1}{2}(\dot{\upsilon}_{idv})^2 + \frac{1}{2\beta_{iv}}(\dot{d}_{iv})^2 + \frac{\delta_{iv}}{2\beta_{iv}}d_{iv}^2 + \frac{1}{2}$.

Finally, considering the third equation in (6), we can get $\dot{e}_{i23} = \dot{r}_i - \dot{\upsilon}_{idr}^d$. The following Lyapunov function candidate is defined as $V_{e_{i23}} = \frac{1}{2}m_{33}e_{i23}^2 + \frac{1}{2}m_{33}\tilde{r}_i^2 + \frac{1}{2\beta_{ir}}\tilde{d}_{ir}^2 + \frac{1}{2}y_{idr}^2$

where $\tilde{d}_{ir} = d_{ir} - \hat{d}_{ir}$ and $\beta_{ir} > 0$. Using (5) and (7), the dynamics of $V_{e_{i23}}$ are

$$
\begin{aligned}
\dot{V}_{e_{i23}} &\leq e_{i23}(-d_{32}v_i - d_{33}r_i + \tau_{ir} + \hat{d}_{ir} - m_{33}\dot{v}_{idr}^d + d_i e_{i13}) \\
&- k_{ir1}\tilde{r}_i \tanh(\frac{\tilde{r}_i}{\Xi_{\tilde{r}_i}}) - k_{ir2}\tilde{r}_i^2 + \frac{1}{\beta_{ir}}\tilde{d}_{ir}\dot{d}_{ir} + \frac{1}{\beta_{ir}}\tilde{d}_{ir}(\beta_{ir}(e_{i23} \\
&+ \tilde{r}_i) - \dot{\hat{d}}_{ir}) - (k_{idr2} - \frac{1}{2})y_{ide}^2 - k_{idr1}y_{idr}\tanh(\frac{y_{idr}}{\Xi_{y_{idr}}}) \\
&+ \frac{1}{2}(\dot{v}_{idr})^2 - e_{i23}\sum_{j=1}^{M} a_{ij}e_{i13} \quad (28)
\end{aligned}
$$

Subsequently, the force $\tau_{ir}$ in yaw is designed as follows:

$$
\begin{aligned}
\tau_{ir} &= -k_{i21r}\tanh(\frac{e_{i23}}{\Xi_{e_{i23}}}) - k_{i22r}e_{i23} + d_{32}v_i + d_{33}r_i \\
&- \hat{d}_{ir} + m_{33}\dot{v}_{idr}^d - \sum_{j=1}^{M} a_{ij}e_{i13} \quad (29)
\end{aligned}
$$

where $k_{i21r} > 0$ and $k_{i22r} > 0$.

Substituting (4) and (29) into (28), we have

$$
\begin{aligned}
\dot{V}_{e_{i23}} &\leq -k_{i21r}e_{i23}\tanh(\frac{e_{i23}}{\Xi_{e_{i23}}}) - k_{i22r}e_{i23}^2 \\
&- k_{ir1}\tilde{r}_i \tanh(\frac{\tilde{r}_i}{\Xi_{\tilde{r}_i}}) - k_{ir2}\tilde{r}_i^2 + \frac{1}{\beta_{ir}}\tilde{d}_{ir}\dot{d}_{ir} \\
&+ \frac{\delta_{ir}}{\beta_{ir}}\tilde{d}_{ir}\hat{d}_{ir} - k_{idr1}y_{idr}\tanh(\frac{y_{idr}}{\Xi_{y_{idr}}}) \\
&- (k_{idr2} - \frac{1}{2})y_{idr}^2 + \frac{1}{2}(\dot{v}_{idr})^2 - e_{i23}\cdot\sum_{j=1}^{M} a_{ij}e_{i13} \quad (30)
\end{aligned}
$$

Utilizing Lemma 3 and Young's inequality, we get

$$
\begin{aligned}
-k_{i21r}e_{i23}\tanh(\frac{e_{i23}}{\Xi_{e_{i23}}}) &\leq -k_{i21r}(e_{i23}^2)^{\frac{1}{2}} + k_{i21r}\kappa\Xi_{e_{i23}} \\
-k_{ir1}\tilde{r}_i \tanh(\frac{\tilde{r}_i}{\Xi_{\tilde{r}_i}}) &\leq -k_{ir1}(\tilde{r}_i^2)^{\frac{1}{2}} + k_{ir1}\kappa\Xi_{\tilde{r}_i} \\
-k_{idr1}y_{idr}\tanh(\frac{y_{idr}}{\Xi_{y_{idr}}}) &\leq -k_{idr1}(y_{idr}^2)^{\frac{1}{2}} + k_{idr1}\kappa\Xi_{y_{idr}} \\
\frac{1}{\beta_{ir}}\tilde{d}_{ir}\dot{d}_{ir} + \frac{\delta_{ir}}{\beta_{ir}}\tilde{d}_{ir}\hat{d}_{ir} &\leq -(\frac{1}{2\beta_{ir}}\tilde{d}_{ir}^2)^{\frac{1}{2}} - \frac{\delta_{ir}-2}{2\beta_{ir}}\tilde{d}_{ir}^2 \\
&+ \frac{1}{2\beta_{ir}}(\dot{d}_{ir})^2 + \frac{\delta_{ir}}{2\beta_{ir}}d_{ir}^2 \\
&+ \frac{1}{2} \quad (31)
\end{aligned}
$$

Combining (30) and (31), we have

$$
\begin{aligned}
\dot{V}_{e_{i23}} &\leq -k_{i21r}(e_{i23}^2)^{\frac{1}{2}} - k_{i22r}e_{i23}^2 - k_{ir1}(\tilde{r}_i^2)^{\frac{1}{2}} - k_{ir2}\tilde{r}_i^2 \\
&- (\frac{1}{2\beta_{ir}}\tilde{d}_{ir}^2)^{\frac{1}{2}} - \frac{\delta_{ir}-2}{2\beta_{ir}}\tilde{d}_{ir}^2 - k_{idr1}(y_{idr}^2)^{\frac{1}{2}} - (k_{idr2} \\
&- \frac{1}{2})y_{idr}^2 - e_{i23}\cdot d_i e_{i13} + T_{e_{i23}} \quad (32)
\end{aligned}
$$

where $T_{e_{i23}} = k_{i21r}\kappa\Xi_{e_{i23}} + k_{ir1}\kappa\Xi_{\tilde{r}_i} + k_{idr1}\kappa\Xi_{y_{idr}} + \frac{1}{2}(\dot{v}_{idr})^2 + \frac{1}{2\beta_{ir}}(\dot{d}_{ir})^2 + \frac{\delta_{ir}}{2\beta_{ir}}d_{ir}^2 + \frac{1}{2}$.

## C. Stability analysis

**Theorem 1**: Consider the networked USs (1) under the distributed controllers (19), (24), (29), the first-order filters (7) and the disturbance estimators (4). If the initial conditions are bounded with $e_{i11}(0) < v_{ix}(0)$, $e_{i12}(0) < v_{iy}(0)$ and $e_{i13}(0) < v_{i\psi}(0)(i = 1, 2, \ldots, N)$, the following properties are true: *i)* All the state signals and disturbance estimation errors are bounded in finite time. *ii)* The synchronization tracking errors can be included in the specific decaying ranges.

**Proof**: The Lyapunov function is given as

$$
\begin{aligned}
V &= \sum_{i=1}^{N}(\frac{1}{2}\ln\frac{v_{ix}^2(t)}{v_{ix}^2(t)-e_{i11}^2} + \frac{1}{2}\ln\frac{v_{iy}^2(t)}{v_{iy}^2(t)-e_{i12}^2} \\
&+ \frac{1}{2}\ln\frac{v_{i\psi}^2(t)}{v_{i\psi}^2(t)-e_{i13}^2} + \frac{1}{2}m_{11}e_{i21}^2 + \frac{1}{2}m_{11}\tilde{u}_i^2 + \frac{1}{2\beta_{iu}}\tilde{d}_{iu}^2 \\
&+ \frac{1}{2}y_{idu}^2 + \frac{1}{2}m_{22}e_{i22}^2 + \frac{1}{2}m_{22}\tilde{v}_i^2 + \frac{1}{2\beta_{iv}}\tilde{d}_{iv}^2 \\
&+ \frac{1}{2}y_{idv}^2 + \frac{1}{2}m_{33}e_{i23}^2 + \frac{1}{2}m_{33}\tilde{r}_i^2 + \frac{1}{2\beta_{ir}}\tilde{d}_{ir}^2 + \frac{1}{2}y_{idr}^2)(33)
\end{aligned}
$$

Taking the time derivative of $V$ yields

$$
\begin{aligned}
\dot{V} &= \sum_{i=1}^{N}(-k_{i1x}\frac{e_{i11}^2}{v_{ix}^2(t)-e_{i11}^2} + \frac{\dot{v}_{ix}(t)}{v_{ix}(t)} + (1 \\
&- 2\tanh^2(\frac{e_{i11}}{\bar{v}_{ix}}))(-\frac{v_{ix}(t)\dot{v}_{ix}(t)}{v_{ix}^2(t)-e_{i11}^2}) \\
&- k_{i1y}\frac{e_{i12}^2}{v_{iy}^2(t)-e_{i12}^2} + \frac{\dot{v}_{iy}(t)}{v_{iy}(t)} + (1 \\
&- 2\tanh^2(\frac{e_{i12}}{\bar{v}_{iy}}))(-\frac{v_{iy}(t)\dot{v}_{iy}(t)}{v_{iy}^2(t)-e_{i12}^2}) \\
&- k_{i1\psi}\frac{e_{i13}^2}{v_{i\psi}^2(t)-e_{i13}^2} + \frac{\dot{v}_{i\psi}(t)}{v_{i\psi}(t)} \\
&+ (1 - 2\tanh^2(\frac{e_{i13}}{\bar{v}_{i\psi}}))(-\frac{v_{i\psi}(t)\dot{v}_{i\psi}(t)}{v_{i\psi}^2(t)-e_{i13}^2}) \\
&+ d_i^2 y_{idu}^2 + d_i^2 y_{idv}^2 + \frac{1}{2}d_i^2 y_{idr}^2 \\
&- k_{i21u}(e_{i21}^2)^{\frac{1}{2}} - k_{i22u}e_{i21}^2 - k_{iu1}(\tilde{u}_i^2)^{\frac{1}{2}} \\
&- k_{iu2}\tilde{u}_i^2 - (\frac{1}{2\beta_{iu}}\tilde{d}_{iu}^2)^{\frac{1}{2}} - \frac{\delta_{iu}-2}{2\beta_{iu}}\tilde{d}_{iu}^2 \\
&- k_{idu1}(y_{idu}^2)^{\frac{1}{2}} - (k_{idu2} - \frac{1}{2})y_{idu}^2 + T_{ei21} \\
&- k_{i21v}(e_{i22}^2)^{\frac{1}{2}} - k_{i22v}e_{i22}^2 - k_{iv1}(\tilde{v}_i^2)^{\frac{1}{2}} \\
&- k_{iv2}\tilde{v}_i^2 - (\frac{1}{2\beta_{iv}}\tilde{d}_{iv}^2)^{\frac{1}{2}} - \frac{\delta_{iv}-2}{2\beta_{iv}}\tilde{d}_{iv}^2 \\
&- k_{idv1}(y_{idv}^2)^{\frac{1}{2}} - (k_{idv2} - \frac{1}{2})y_{idv}^2 + T_{ei22} \\
&- k_{i21r}(e_{i23}^2)^{\frac{1}{2}} - k_{i22r}e_{i23}^2 - k_{ir1}(\tilde{r}_i^2)^{\frac{1}{2}} \\
&- k_{ir2}\tilde{r}_i^2 - (\frac{1}{2\beta_{ir}}\tilde{d}_{ir}^2)^{\frac{1}{2}} - \frac{\delta_{ir}-2}{2\beta_{ir}}\tilde{d}_{ir}^2 \\
&- k_{idr1}(y_{idr}^2)^{\frac{1}{2}} - (k_{idr2} - \frac{1}{2})y_{idr}^2 + T_{ei23}) \quad (34)
\end{aligned}
$$

By utilizing Lemma 5, we have

$$-k_{i1x}\frac{e_{i11}^2}{v_{ix}^2(t)-e_{i11}^2} \leq -(k_{i1x}-\frac{1}{2})\ln\frac{v_{ix}^2(t)}{v_{ix}^2(t)-e_{i11}^2}$$
$$-(\ln\frac{v_{ix}^2(t)}{v_{ix}^2(t)-e_{i11}^2})^{\frac{1}{2}}+\frac{1}{2}$$
$$-k_{i1y}\frac{e_{i12}^2}{v_{iy}^2(t)-e_{i12}^2} \leq -(k_{i1y}-\frac{1}{2})\ln\frac{v_{iy}^2(t)}{v_{iy}^2(t)-e_{i12}^2}$$
$$-(\ln\frac{v_{iy}^2(t)}{v_{iy}^2(t)-e_{i12}^2})^{\frac{1}{2}}+\frac{1}{2}$$
$$-k_{i1\psi}\frac{e_{i13}^2}{v_{i\psi}^2(t)-e_{i13}^2} \leq -(k_{i1\psi}-\frac{1}{2})\ln\frac{v_{i\psi}^2(t)}{v_{i\psi}^2(t)-e_{i13}^2}$$
$$-(\ln\frac{v_{i\psi}^2(t)}{v_{i\psi}^2(t)-e_{i13}^2})^{\frac{1}{2}}+\frac{1}{2} \quad (35)$$

Furthermore, by the definition of $v_{ix}(t)$, $v_{iy}(t)$ and $v_{i\psi}(t)$, $v_{ix}(t)$, $\dot{v}_{ix}(t)$, $v_{iy}(t)$, $\dot{v}_{iy}(t)$, $v_{i\psi}(t)$ and $\dot{v}_{i\psi}(t)$ are bounded. Then, the following results are true.

$$\frac{\dot{v}_{ix}(t)}{v_{ix}(t)} \leq K_{1i}, \frac{\dot{v}_{iy}(t)}{v_{iy}(t)} \leq K_{2i}, \frac{\dot{v}_{i\psi}(t)}{v_{i\psi}(t)} \leq K_{3i} \quad (36)$$

where $K_{1i}>0$, $K_{2i}>0$ and $K_{3i}>0$.

Subsequently, substituting (35) and (36) into (34) gets following result:

$$\dot{V}=\sum_{i=1}^{N}(-(k_{i1x}-\frac{1}{2})\ln\frac{v_{ix}^2(t)}{v_{ix}^2(t)-e_{i11}^2}-(\ln\frac{v_{ix}^2(t)}{v_{ix}^2(t)-e_{i11}^2})^{\frac{1}{2}}$$
$$-(k_{i1y}-\frac{1}{2})\ln\frac{v_{iy}^2(t)}{v_{iy}^2(t)-e_{i12}^2}-(\ln\frac{v_{iy}^2(t)}{v_{iy}^2(t)-e_{i12}^2})^{\frac{1}{2}}$$
$$-(k_{i1\psi}-\frac{1}{2})\ln\frac{v_{i\psi}^2(t)}{v_{i\psi}^2(t)-e_{i13}^2}-(\ln\frac{v_{i\psi}^2(t)}{v_{i\psi}^2(t)-e_{i13}^2})^{\frac{1}{2}}$$
$$-k_{i21u}(e_{i21}^2)^{\frac{1}{2}}-k_{i22u}e_{i21}^2-k_{iu1}(\tilde{u}_i^2)^{\frac{1}{2}}-k_{iu2}\tilde{u}_i^2$$
$$-(\frac{1}{2\beta_{iu}}\tilde{d}_{iu}^2)^{\frac{1}{2}}-\frac{\delta_{iu}-2}{2\beta_{iu}}\tilde{d}_{iu}^2-k_{idu1}(y_{idu}^2)^{\frac{1}{2}}-(k_{idu2}$$
$$-\frac{1}{2}-d_i^2)y_{idu}^2-k_{i21v}(e_{i22}^2)^{\frac{1}{2}}-k_{i22v}e_{i22}^2-k_{iv1}(\tilde{v}_i^2)^{\frac{1}{2}}$$
$$-k_{iv2}\tilde{v}_i^2-(\frac{1}{2\beta_{iv}}\tilde{d}_{iv}^2)^{\frac{1}{2}}-\frac{\delta_{iv}-2}{2\beta_{iv}}\tilde{d}_{iv}^2-k_{idv1}(y_{idv}^2)^{\frac{1}{2}}$$
$$-(k_{idv2}-\frac{1}{2}-d_i^2)y_{idv}^2-k_{i21r}(e_{i23}^2)^{\frac{1}{2}}-k_{i22r}e_{i23}^2$$
$$-k_{ir1}(\tilde{r}_i^2)^{\frac{1}{2}}-k_{ir2}\tilde{r}_i^2-(\frac{1}{2\beta_{ir}}\tilde{d}_{ir}^2)^{\frac{1}{2}}-\frac{\delta_{ir}-2}{2\beta_{ir}}\tilde{d}_{ir}^2$$
$$-k_{idr1}(y_{idr}^2)^{\frac{1}{2}}-(k_{idr2}-\frac{1}{2}-\frac{d_i^2}{2})y_{idr}^2+\frac{3}{2}+K_{1i}$$
$$+K_{2i}+K_{3i}+T_{ei21}+T_{ei22}+T_{ei23}$$
$$+(1-2\tanh^2(\frac{e_{i11}}{\bar{v}_{ix}}))(-\frac{v_{ix}(t)\dot{v}_{ix}(t)}{v_{ix}^2(t)-e_{i11}^2})$$
$$+(1-2\tanh^2(\frac{e_{i12}}{\bar{v}_{iy}}))(-\frac{v_{iy}(t)\dot{v}_{iy}(t)}{v_{iy}^2(t)-e_{i12}^2})$$
$$+(1-2\tanh^2(\frac{e_{i13}}{\bar{v}_{i\psi}}))(-\frac{v_{i\psi}(t)\dot{v}_{i\psi}(t)}{v_{i\psi}^2(t)-e_{i13}^2})) \quad (37)$$

Defining $\lambda_{1i}=\min\{k_{i1x}-\frac{1}{2},k_{i1y}-\frac{1}{2},k_{i1\psi}-\frac{1}{2},k_{i22u},$
$k_{iu2},\frac{\delta_{iu}-2}{2\beta_{iu}},k_{idu2}-\frac{1}{2}-d_i^2,k_{i22v},k_{iv2},\frac{\delta_{iv}-2}{2\beta_{iv}},k_{idv2}-\frac{1}{2}-d_i^2,$
$k_{i22r},k_{ir2},\frac{\delta_{ir}-2}{2\beta_{ir}},k_{idr2}-\frac{1}{2}-\frac{d_i^2}{2}\}$, $\lambda_1=\min\{\lambda_{11},\lambda_{12},\ldots,$
$\lambda_{1N}\}$, $\lambda_{2i}=\min\{1,k_{i21u},k_{iu1},k_{idu1},k_{i21v},k_{iv1},k_{idv1},k_{i21r},$
$k_{ir1},k_{idr1}\}$ and $\lambda_2=\min\{\lambda_{21},\lambda_{22},\ldots,\lambda_{2N}\}$, we can get

$$\dot{V}\leq-\lambda_1V-\lambda_2V^{\frac{1}{2}}+\sum_{i=1}^{N}(\frac{3}{2}+K_{1i}+K_{2i}+K_{3i}+T_{ei21}$$
$$+T_{ei22}+T_{ei23}+(1-2\tanh^2(\frac{e_{i11}}{\bar{v}_{ix}}))(-\frac{v_{ix}(t)\dot{v}_{ix}(t)}{v_{ix}^2(t)-e_{i11}^2})$$
$$+(1-2\tanh^2(\frac{e_{i12}}{\bar{v}_{iy}}))(-\frac{v_{iy}(t)\dot{v}_{iy}(t)}{v_{iy}^2(t)-e_{i12}^2})$$
$$+(1-2\tanh^2(\frac{e_{i13}}{\bar{v}_{i\psi}}))(-\frac{v_{i\psi}(t)\dot{v}_{i\psi}(t)}{v_{i\psi}^2(t)-e_{i13}^2})) \quad (38)$$

In (38), according to Lemma 4, $e_{i11}$, $e_{i12}$ and $e_{i13}$ respectively determine the signs of $1-2\tanh^2(\frac{e_{i11}}{\bar{v}_{ix}})$, $1-2\tanh^2(\frac{e_{i12}}{\bar{v}_{iy}})$ and $1-2\tanh^2(\frac{e_{i13}}{\bar{v}_{i\psi}})$. Therefore, the following three cases will be given.

*Case 1*: For $\bar{v}_{ix}$, $\bar{v}_{iy}$ and $\bar{v}_{i\psi}$, we have $e_{i11}\in\Omega_{e_{i11}}=\{e_{i11}||e_{i11}|<0.8814\bar{v}_{ix}\}$, $e_{i12}\in\Omega_{e_{i12}}=\{e_{i12}||e_{i12}|<0.8814\bar{v}_{iy}\}$ and $e_{i13}\in\Omega_{e_{i13}}=\{e_{i13}||e_{i13}|<0.8814\bar{v}_{i\psi}\}$. Because $e_{i11}$, $v_{ix}(t)$, $\dot{v}_{ix}(t)$, $e_{i12}$, $v_{iy}(t)$, $\dot{v}_{iy}(t)$, $e_{i13}$, $v_{i\psi}(t)$ and $\dot{v}_{i\psi}(t)$ are bounded, $(1-2\tanh^2(\frac{e_{i11}}{\bar{v}_{ix}}))(-\frac{v_{ix}(t)\dot{v}_{ix}(t)}{v_{ix}^2(t)-e_{i11}^2})+(1-2\tanh^2(\frac{e_{i12}}{\bar{v}_{iy}}))(-\frac{v_{iy}(t)\dot{v}_{iy}(t)}{v_{iy}^2(t)-e_{i12}^2})+(1-2\tanh^2(\frac{e_{i13}}{\bar{v}_{i\psi}}))(-\frac{v_{i\psi}(t)\dot{v}_{i\psi}(t)}{v_{i\psi}^2(t)-e_{i13}^2})$ is also bounded and it's supposed that $c_i(>0)$ is the upper boundedness. Then, for (38), we can obtain $\dot{V}\leq-\lambda_1V-\lambda_2V^{\frac{1}{2}}+d$ where $d=\Sigma_{i=1}^{N}(\frac{3}{2}+K_{1i}+K_{2i}+K_{3i}+T_{ei21}+T_{ei22}+T_{ei23}+c_i)$.

*Case 2*: For $\bar{v}_{ix}$, $\bar{v}_{iy}$ and $\bar{v}_{i\psi}$, we have $e_{i11}\notin\Omega_{e_{i11}}=\{e_{i11}||e_{i11}|<0.8814\bar{v}_{ix}\}$, $e_{i12}\notin\Omega_{e_{i12}}=\{e_{i12}||e_{i12}|<0.8814\bar{v}_{iy}\}$ and $e_{i13}\notin\Omega_{e_{i13}}=\{e_{i13}||e_{i13}|<0.8814\bar{v}_{i\psi}\}$. According to Lemma 4, based on the fact that $-\frac{v_{ix}(t)\dot{v}_{ix}(t)}{v_{ix}^2(t)-e_{i11}^2}\geq0$, $-\frac{v_{iy}(t)\dot{v}_{iy}(t)}{v_{iy}^2(t)-e_{i12}^2}\geq0$ and $-\frac{v_{i\psi}(t)\dot{v}_{i\psi}(t)}{v_{i\psi}^2(t)-e_{i13}^2}\geq0$, we can obtain $(1-2\tanh^2(\frac{e_{i11}}{\bar{v}_{ix}}))(-\frac{v_{ix}(t)\dot{v}_{ix}(t)}{v_{ix}^2(t)-e_{i11}^2})+(1-2\tanh^2(\frac{e_{i12}}{\bar{v}_{iy}}))(-\frac{v_{iy}(t)\dot{v}_{iy}(t)}{v_{iy}^2(t)-e_{i12}^2})+(1-2\tanh^2(\frac{e_{i13}}{\bar{v}_{i\psi}}))(-\frac{v_{i\psi}(t)\dot{v}_{i\psi}(t)}{v_{i\psi}^2(t)-e_{i13}^2})\leq0$. Then, for (38), we can obtain $\dot{V}\leq-\lambda_1V-\lambda_2V^{\frac{1}{2}}+d$ where $d=\Sigma_{i=1}^{N}(\frac{3}{2}+K_{1i}+K_{2i}+K_{3i}+T_{ei21}+T_{ei22}+T_{ei23}+c_i)$.

*Case 3*: For $e_{i11}$, $e_{i12}$ and $e_{i13}$, some are inside the set and some are outside the set. Based on Case 1 and Case 2, we can obtain $\dot{V}\leq-\lambda_1V-\lambda_2V^{\frac{1}{2}}+d$ where $d=\Sigma_{i=1}^{N}(\frac{3}{2}+K_{1i}+K_{2i}+K_{3i}+T_{ei21}+T_{ei22}+T_{ei23}+c_i)$.

According to the above three cases, (38) can be rewritten as

$$\dot{V}\leq-\lambda_1V-\lambda_2V^{\frac{1}{2}}+d \quad (39)$$

According to Lemma 1, we can get

$$0\leq V\leq\min\{\frac{d}{(1-\theta_0)\lambda_1},(\frac{d}{(1-\theta_0)\lambda_2})^{\frac{1}{2}}\},t>T_r$$

where $0 < \theta_0 < 1$ and

$$T_r \leq \max\{\frac{2}{\theta_0 \lambda_1} \ln \frac{\theta_0 \lambda_1 \sqrt{V(0)} + \lambda_2}{\lambda_2},$$
$$\frac{2}{\lambda_1} \ln \frac{\lambda_1 \sqrt{V(0)} + \theta_0 \lambda_2}{\theta_0 \lambda_2}\} \qquad (40)$$

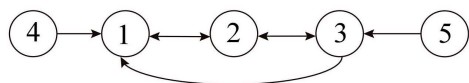

Fig. 1. Topology of communication graph.

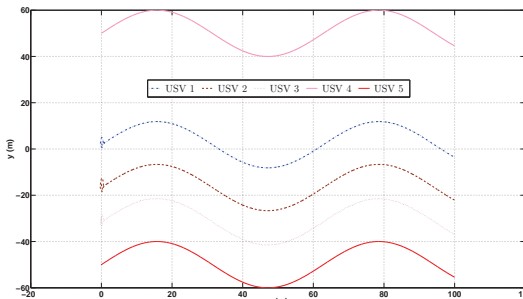

Fig. 2. Trajectories of US♯1-US♯5.

Then, we can obtain that, according to Lemma 4, all the state signals and disturbance estimation errors are bounded when $t > T_r$.

By (39), it can be concluded that $\ln \frac{v_{ix}^2(t)}{v_{ix}^2(t) - e_{i11}^2} \leq 2\min\{\frac{d}{(1-\theta_0)\lambda_1}, (\frac{d}{(1-\theta_0)\lambda_2})^2\}$, $\ln \frac{v_{iy}^2(t)}{v_{iy}^2(t) - e_{i12}^2} \leq 2\min\{\frac{d}{(1-\theta_0)\lambda_1}, (\frac{d}{(1-\theta_0)\lambda_2})^2\}$ and $\ln \frac{v_{i\psi}^2(t)}{v_{i\psi}^2(t) - e_{i13}^2} \leq 2\min\{\frac{d}{(1-\theta_0)\lambda_1}, (\frac{d}{(1-\theta_0)\lambda_2})^2\}$.

Subsequently, we know that $\frac{v_{ix}^2(t)}{v_{ix}^2(t) - e_{i11}^2} \leq e^{2\min\{\frac{d}{(1-\theta_0)\lambda_1}, (\frac{d}{(1-\theta_0)\lambda_2})^2\}}$, $\frac{v_{iy}^2(t)}{v_{iy}^2(t) - e_{i12}^2} \leq e^{2\min\{\frac{d}{(1-\theta_0)\lambda_1}, (\frac{d}{(1-\theta_0)\lambda_2})^2\}}$ and $\frac{v_{i\psi}^2(t)}{v_{i\psi}^2(t) - e_{i13}^2} \leq e^{2\min\{\frac{d}{(1-\theta_0)\lambda_1}, (\frac{d}{(1-\theta_0)\lambda_2})^2\}}$.

Because $v_{ix}^2(t) - e_{i11}^2 > 0$, $v_{iy}^2(t) - e_{i12}^2 > 0$ and $v_{i\psi}^2(t) - e_{i13}^2 > 0$, we have $|e_{i11}| \leq \sqrt{1 - \frac{1}{e^{2\min\{\frac{d}{(1-\theta_0)\lambda_1}, (\frac{d}{(1-\theta_0)\lambda_2})^2\}}}} |v_{ix}(t)|$, $|e_{i12}| \leq \sqrt{1 - \frac{1}{e^{2\min\{\frac{d}{(1-\theta_0)\lambda_1}, (\frac{d}{(1-\theta_0)\lambda_2})^2\}}}} |v_{iy}(t)|$ and $|e_{i13}| \leq \sqrt{1 - \frac{1}{e^{2\min\{\frac{d}{(1-\theta_0)\lambda_1}, (\frac{d}{(1-\theta_0)\lambda_2})^2\}}}} |v_{i\psi}(t)|$ which imply that $|e_{i11}| \leq |v_{ix}(t)|, |e_{i12}| \leq |v_{iy}(t)|, |e_{i13}| \leq |v_{i\psi}(t)|$ where $\min\{\frac{d}{(1-\theta_0)\lambda_1}, (\frac{d}{(1-\theta_0)\lambda_2})^2\} > 0$.

The proof is thus completed.

## IV. SIMULATION RESULTS

For the utilizability of the obtained method, 5 USs including 3 follower USs (US♯1-US♯3) and 2 leader USs (US♯4-US♯5) are given. The topological structure is given in Fig. 1. The adjacency matrix is selected as $A =$ $[0, 0.4, 0.5, 0.5, 0; 0.4, 0, 0.5, 0, 0; 0, 0.5, 0, 0, 0.4; 0, 0, 0, 0, 0; 0, 0, 0, 0, 0]$.

The disturbance is

$$d_i = [100000\sin(0.1t), 100000\cos(t), 100000\sin(t)]^T$$

The structure parameters of the follower US♯1-US♯3 are $M = [5.3122 \times 10^6, 0, 0; 0, 8.2831 \times 10^6, 0; 0, 0, 3.7454 \times 10^9]$ and $D = [5.0242 \times 10^4, 0, 0; 0, 2.7229 \times 10^5, -4.3933 \times 10^6; 0, -4.3933 \times 10^6, 4.1894 \times 10^8]$.

In this paper, the trajectories from the leader US♯4-US♯5 are $\eta_{4d} = [0.5t; 10\sin(0.05t) + 50; \arctan(\cos(0.05t))]$ and $\eta_{5d} = [0.5t, 10\sin(0.05t) - 50, \arctan(\cos(0.05t))]$.

The follower USs' initial values are

$$\eta_1(0) = [0\text{m}, 3\text{m}, 0.5\text{rad}]^T$$
$$\upsilon_1(0) = [0\text{m/s}, 0\text{m/s}, 0\text{rad/s}]^T$$
$$\eta_2(0) = [0\text{m}, -16\text{m}, 0.7854\text{rad}]^T$$
$$\upsilon_2(0) = [0\text{m/s}, 0\text{m/s}, 0\text{rad/s}]^T$$
$$\eta_3(0) = [0\text{m}, -30\text{m}, 0.7854\text{rad}]^T$$
$$\upsilon_3(0) = [0\text{m/s}, 0\text{m/s}, 0\text{rad/s}]^T$$

The controller gains are chosen as $k_{i1x} = k_{i1y} = k_{i1\psi} = 5$, $\bar{v}_{ix} = \bar{v}_{iy} = \bar{v}_{i\psi} = 1$, $k_{i21u} = k_{i21v} = k_{i21r} = 2$, $k_{i22u} = k_{i22v} = k_{i22r} = 5$ and $\Xi_{e_{i21}} = \Xi_{e_{i22}} = \Xi_{e_{i23}} = 0.1$. The disturbance estimators' initial states are $\hat{v}_i(0) = [0, 0, 0]^T$ and the parameters are chosen as $k_{iu1} = k_{iu2} = k_{iv1} = k_{iv2} = k_{ir1} = k_{ir2} = 1$ and $\Xi_{\tilde{u}_i} = \Xi_{\tilde{v}_i} = \Xi_{\tilde{r}_i} = 0.1$. The design parameters of first-order filters are $k_{idu1} = k_{idv1} = k_{idr1} = 2$, $k_{idu2} = k_{idv2} = k_{idr2} = 15$ and $\Xi_{y_{idu}} = \Xi_{y_{idv}} = \Xi_{y_{idr}} = 0.1$. The performance functions for the PPC are $v_{ix}(t) = v_{iy}(t) = v_{i\psi}(t) = (10 - 2)e^{-20t} + 2$.

Fig. 2 shows the curves of US♯1-US♯5. It is obtained that the curves of US♯1-US♯3 are included in the convex hull generated by the trajectories of US♯4-US♯5. From Fig. 3, the synchronization errors of US♯1-US♯3 are included in the specified performance bounds.

The velocity curves of the follower US♯1-US♯3 are shown in Fig. 4. Fig. 5 gives the controller responses of US♯1-US♯3. From Fig. 4 and Fig. 5, the velocities and control inputs are bounded. Fig. 6 shows the disturbance estimation errors, which are finite-time bounded.

## V. CONCLUSION

The article has discussed the distributed security control issue for USs with the disturbances. To constrain the tracking errors within predefined boundaries, infinite-time performance functions have been introduced and the tracking errors were successfully controlled within the prespecified ranges. The disturbances were approximated by the disturbance estimators with finite-time estimation errors. Compared with the cooperative control of USs with one leader US, the obtained controller can solve the distributed security control issue with multiple leader USs and make further tracking errors satisfy the transient and steady-state performances. Numerical simulation

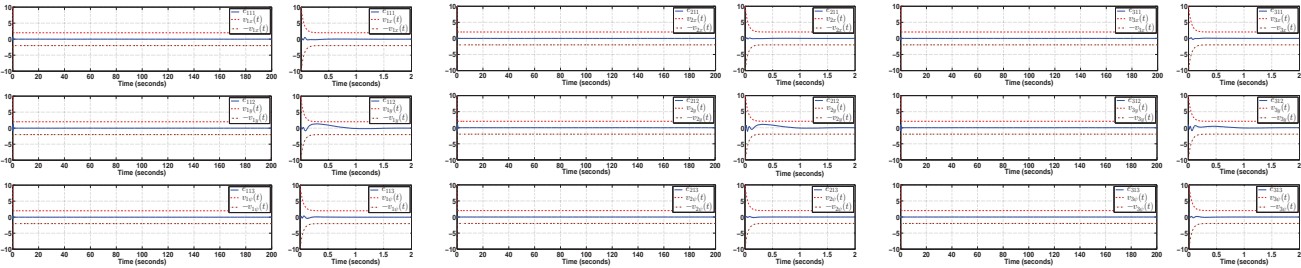

(a) Synchronization tracking errors of USV♯1.    (b) Synchronization tracking errors of USV♯2.    (c) Synchronization tracking errors of USV♯3.

Fig. 3. Synchronization tracking errors of US♯1-US♯3.

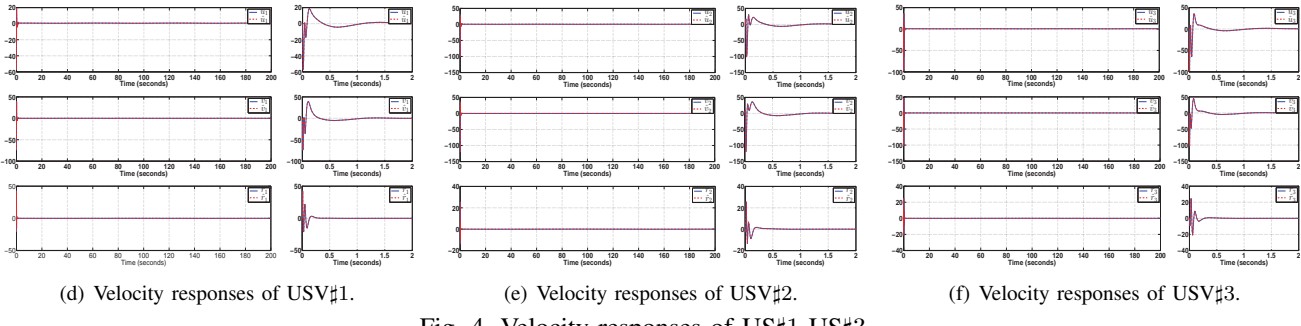

(d) Velocity responses of USV♯1.    (e) Velocity responses of USV♯2.    (f) Velocity responses of USV♯3.

Fig. 4. Velocity responses of US♯1-US♯3.

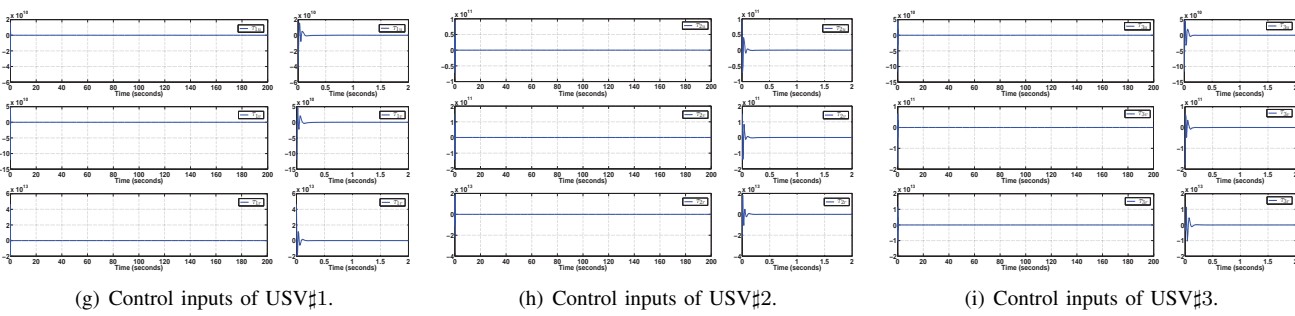

(g) Control inputs of USV♯1.    (h) Control inputs of USV♯2.    (i) Control inputs of USV♯3.

Fig. 5. Control inputs of US♯1-US♯3.

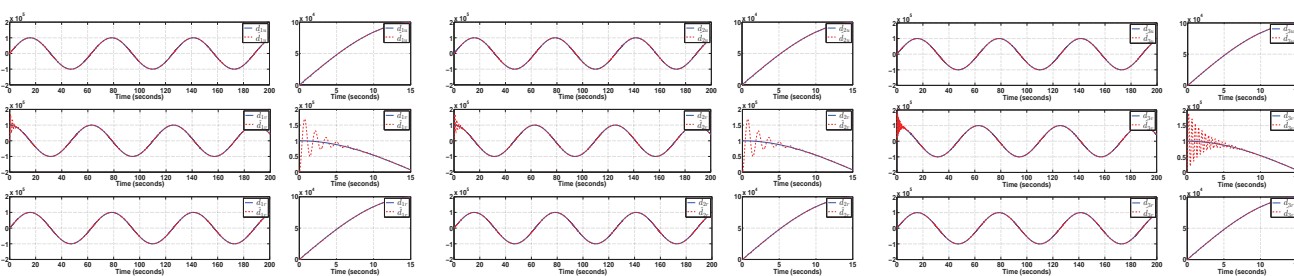

(j) External disturbance estimations of USV♯1.    (k) External disturbance estimations of USV♯2.    (l) External disturbance estimations of USV♯3.

Fig. 6. External disturbance estimations of US♯1-US♯3.

results have revealed that the synchronization tracking errors satisfied the pre-specified prescribed performance.

However, in practice, the considered system states are normally unknown or unmeasurable. Therefore, the issue of how to propose the observer-based containment control scheme can be discussed in our future research.

## ACKNOWLEDGMENT

This research was supported by the National Natural Science Foundation of China under Grant No. 51939001, No. 61976033 and No. 62173046.

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
