# OpenReview forum: "Distributed Finite-time Prescribed Performance Security Control for Unmanned Ships Utilizing the Novel Disturbance Estimator"
_IEEE.org/ICIST/2024/Conference — IEEE ICIST 2024 Conference Submission_

### Official Review · Reviewer_nxbT · 2024-08-25
**minor repair**

**Rating:** 9
**Confidence:** 3

**Review:**

1. Notations in this paper are easily confused, especially the symbols of damping matrix and D = diag[d1, d2, . . . , dM].
2. When using abbreviations for the first time, the full name should be provided, such as "USVs".
3. The simulation section should provide more selection parameters to facilitate readers' understanding.

---

### Official Review · Reviewer_zQum · 2024-08-27
**The topic under consideration is interesting. This paper can be accepted after minor modifications.**

**Rating:** 9
**Confidence:** 3

**Review:**

This paper developed a distributed security control protocol of unmanned ships with the disturbances. The topic under consideration is interesting. Detailed comments and suggestions are listed as follows.

1.	In (1), the model of $i$th follower has been defined, but the model of leaders is not be found.
2.	The derivation of (15) cannot be easily followed. Please give a detailed explanation in the content.
3.	The English writing of the paper needs to be further polished, and some typos should be fixed, such as “Take into account the networked USVs (1) under the distributed controllers (19), (24), (29), the first-order filters (7) and the disturbance estimators (4)” in Page 6.  Moreover, the manuscript format should be revised.

---

### Official Review · Reviewer_VWW6 · 2024-08-30
**comment**

**Rating:** 7
**Confidence:** 5

**Review:**

This paper investigates a distributed security control problem for unmanned ships with the disturbances. However, in the reviewer’s opinion, there are some comments in the paper which should be addressed by the authors:
1.	Introduction about the prescribed performance control is limited, it is recommended to add some literature.
2.	The label format of formulas 8 and 9 are not consistent with other formulas and the alignment in the formulas are not standardized, please correct it.
3.	The arrangement of some simulation results is not quite reasonable, please re-typeset.

---

### Decision · Program_Chairs · 2024-09-06

Accept (Oral)